# Rapid Assessment of Anthocyanins Content of Onion Waste through Visible-Near-Short-Wave and Mid-Infrared Spectroscopy Combined with Machine Learning Techniques



Nikolaos Tziolas [1] , Stella A. Ordoudi [2] , Apostolos Tavlaridis [3], Konstantinos Karyotis [1], George Zalidis [1] and Ioannis Mourtzinos [3,*]

[1] Laboratory of Remote Sensing, Spectroscopy, and GIS, Department of Agriculture, Aristotle University of Thessaloniki, 54124 Thessaloniki, Greece; ntziolas@agro.auth.gr (N.T.); kkaryotis@ihu.edu.gr (K.K.); zalidis@agro.auth.gr (G.Z.)

[2] Laboratory of Food Chemistry and Technology, School of Chemistry, Aristotle University of Thessaloniki, 54124 Thessaloniki, Greece; steord@chem.auth.gr

[3] Department of Food Science and Technology, School of Agriculture, Aristotle University of Thessaloniki, 54124 Thessaloniki, Greece; tolistavl@gmail.com

* Correspondence: mourtzinos@agro.auth.gr; Tel.: +302-310-991-637

**Abstract:** A sustainable process for valorization of onion waste would need to entail preliminary sorting out of exhausted or suboptimal material as part of decision-making. In the present study, an approach for monitoring red onion skin (OS) phenolic composition was investigated through Visible Near-Short-Wave infrared (VNIR-SWIR) (350–2500 nm) and Fourier-Transform-Mid-Infrared (FT-MIR) (4000–600 cm$^{-1}$) spectral analyses and Machine-Learning (ML) methods. Our stepwise approach consisted of: (i) chemical analyses to obtain reference values for Total Phenolic Content (TPC) and Total Monomeric Anthocyanin Content (TAC); (ii) spectroscopic analysis and creation of OS spectral libraries; (iii) generation of calibration and validation datasets; (iv) spectral exploratory analysis and regression modeling via several ML algorithms; and (v) model performance evaluation. Among all, the k-nearest neighbors model from 1st derivative VNIR-SWIR spectra at 350–2500 nm resulted promising for the prediction of TAC (R$^2$ = 0.82, RMSE = 0.52 and RPIQ = 3.56). The 2nd derivative FT-MIR spectral fingerprint among 600–900 and 1500–1600 cm$^{-1}$ proved more informative about the inherent phenolic composition of OS. Overall, the diagnostic value and predictive accuracy of our spectral data support the perspective of employing non-destructive spectroscopic tools in real-time quality control of onion waste.

**Keywords:** VNIR-SWIR; FT-MIR; chemometrics; onion solid waste; natural colorant

## 1. Introduction

Huge amounts of onion (*Allium cepa* L) waste, consisting mainly of the skin and inedible outer scales of the bulb, are generated throughout their supply chain from the farm to retail stores and the households. In 2000, more than 450 000 tonnes of onion solid waste (OSW) were produced in Europe [1]; the tonnage is expected to be much higher today with increasing production [2]. OSW can be considered as an environmental problem because it is not suitable for use as organic fertilizer due to the rapid development of phytopathogenic agents, or as a fodder because of its aroma [3].

A possible solution could be the development of a sustainable process to convert this food waste into a raw material for the food industry [1,4]. Regardless of the season, cultivar, or ripening stage, OSW can be a potential source of fibers, fructooligosaccharides, the alk(en)yl cysteine sulfoxides and certain health-promoting phenolic compounds, especially flavonoids [3]. Onion flavonoids are found especially in the skin and outer layers [5], mainly in the form of quercetin aglycone. At least two major types (including quercetin-

3,4'-O-diglucoside (3,4'-Qdg) and quercetin-4'-O-glucoside (4'-Qmg) [3]) conjugate with glucose.

OSW from red-skinned onions is also rich in anthocyanins [6]. Even though anthocyanins comprise a small percentage of onion bulb flavonoids, they are heavily concentrated in the skin and in a single layer of cells in the epidermal tissue, mainly in the form of cyanidin glucosides, esterified with malonic acid [6]. Red onion dry outer layers could, therefore, be a source of natural colorants that can be extracted with green extraction techniques and used as a replacement of synthetic counterparts like carmine [7].

Several novel technological solutions have been proposed to recover these valuable ingredients from OSW [8]. Many studies focus on improving the incorporation of the extracts into novel, biofunctional food products or in the formulation of food supplements [9,10]. The overall cost/sustainability of the biorefinery processes depend on various factors [1,11] that must comply with rational strategies for waste management, such as stabilization and quality. Standards for preliminary sorting to exclude exhausted or sub-optimal waste would be of great value.

Onsite diagnostic assessment of the onion waste content in phenolic compounds is quite challenging. Complexity of supply chains, multi-scaled production, and heterogeneity of the waste composition hinder large-scale operational investigations. Electrochemical sensors are already used in the food industry as sensitive tools for monitoring polyphenol content in certain commodities [12]. Studies show that non-destructive spectroscopic techniques in the visible-near-short-wave and mid-infrared regions combined with powerful chemometric methods may offer cost-effective, rapid, and versatile tools for monitoring the chemical composition of foods. In this context, the use of VNIR-SWIR and FT-MIR spectrometers at all relevant stages across the onion supply chains along with implementation for quality control of thousands of OSW samples generated would inform decision-making about further waste management processes (re-use, re-cycle, valorization etc.). Whether such technology is mature enough for application to routine analysis of OSW is open to question. Artificial intelligence through machine learning (ML) algorithms has revolutionized the predictive performance of current chemometric methods used in the food sector [13]. In the case of onions, a partial least square regression (PLSR) model for the assessment of total phenolic content and total antioxidant activity of phenolic-rich extracts of onion bulbs has been reported to fit well with data extracted from FT-IR spectral features [14]. However, systematic exploitation of spectral analysis and ML algorithms in OSW lags greatly behind. Some years ago, Vincke et al. [15] utilized NIR spectroscopy along with a Partial Least-Squares Discriminant Analysis (PLS-DA) to automatically sort different parts of onion bulbs produced during specific industrial processes. At that time, Wang and Gitaitis [16] highlighted that light-scattering properties of different onion parts in the VNIR region may be further exploited for non-destructive inspection of diseased onions, but they did not employ chemometric tools in their investigation.

The overarching objective of this study was to examine whether chemometric modelling of VNIR-SWIR spectral data of red OS powder, according to their actual content in monomeric anthocyanins, can feasibly be used for predictive purposes. Reference Ultraviolet-visible (UV-Vis) based chemical assays were employed as a first step to assess the content in total phenols and monomeric anthocyanins. Having specified how the spectral signatures in the VNIR-SWIR were to be recorded under standard acquisition protocols, a series of state-of-the-art ML algorithms for regression analysis were deployed using the whole or sub-regions of the VNIR-SWIR spectra as variable inputs. Whether the diagnostic spectral region for anthocyanins might be extended to the mid-infrared spectrum by employing FT-MIR spectroscopy (in attenuated total reflectance (ATR) mode) was also investigated. In that case, an unsupervised ML-based exploratory approach was used to search for interpretable patterns among the OS spectra.

The focus of this study was to document the steps, driven by the current findings, to turn infrared spectroscopy into an operational tool for the assessment of the anthocyanins present in OSW.

## 2. Materials and Methods

The methodological approach consists of five discrete steps: (i) chemical analysis to obtain reference values for Total Phenolic Content (TPC) and Total monomeric anthocyanins (TAC); (ii) spectroscopic analysis (VNIR-SWIR/ATR-FT-MIR), which includes the creation of the dry OS spectral libraries; iii) generation of calibration and validation datasets; (iv) spectral exploratory analysis and regression modeling of VNIR-SWIR spectra where several ML algorithms were evaluated to predict the content of anthocyanins; and (v) evaluation of the performance metrics obtained by ML algorithms. The overall data processing and analysis workflow is illustrated in Figure 1 and detailed descriptions of the different steps are provided in the sections below.

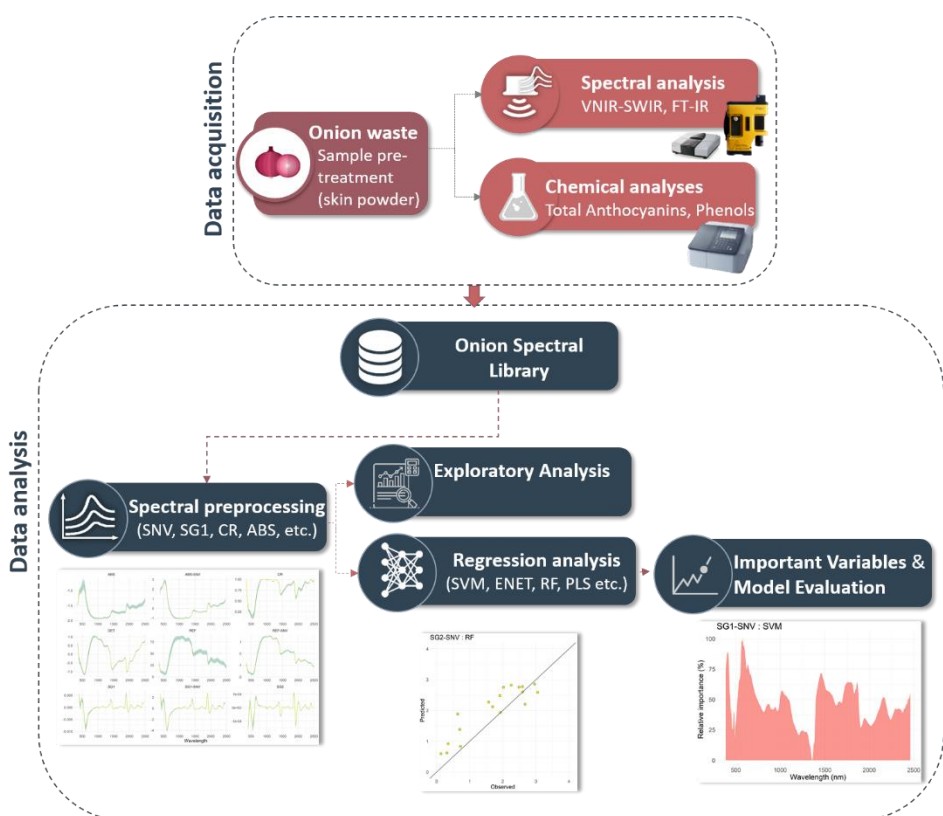

**Figure 1.** Flowchart of the proposed workflow for the onion skin spectral analysis.

### 2.1. Onion Samples (OS)

The bulbs of the *Allium cepa* L. were supplied from various local markets in Greece during the autumn-winter season of 2019 to represent three different retail product lines: one originating from the Netherlands (n = 25) and two from major producing regions in Greece (Thiva, Evritania, n = 13).

The onion skin (OS) test samples were prepared as follows. The bulbs of the red onions were peeled using a sharp blade to remove the outer dry layers and the apical trimmings, which are considered as waste material. Different layers of each sample (dry outer, first inner) were separated. The dry outer and first inner layer skins of each bulb were washed twice in deionized water. The resulting materials were dried at 65 °C for 48 h, ground using a domestic blender (KENWOOD, Havant, UK), powdered in a laboratory mill and then sieved through a 0.5 mm mesh. The material was then mixed to represent 38 test samples of distinct origin. Random combinations of 6 out 38 samples were produced (n = 8) to enhance color variance. The inner and outer layers of 15 individual bulbs from the Netherlands batch were also treated separately (n = 26). In total, 72 OS test samples were used in this study.

*2.2. Chemical Characterization of the OS Extract*

2.2.1. Chemicals

All solvents or analytical standards, such as Folin–Ciocalteu reagent, gallic acid, and sodium chloride ($Na_2CO_3$), were purchased from Sigma-Aldrich, Chemie GmbH (Taufkirchen, Germany).

2.2.2. Preparation of OS Extract

OS powder was mixed with solvent (liquid-to-solid ratio of 10 mL/g), composed of (70% *v/v*) ethanol in water, at pH = 1. The material was subjected to extraction at 25 °C for 15 min in an Ultrasons-H ultrasonic bath (J.P. Selecta Barcelona, Spain). Following extraction, the samples were filtered through a 0.45 μm nylon membrane filters (BGB, USA). The clear supernatant was stored at −20 °C until used for further analysis.

2.2.3. Determination of Total Phenolic Content (TPC)

In brief, 30 μL of all dissolved extracts were mixed, separately, with 2370 μL of deionized water and 150 μL undiluted Folin Ciocalteu's reagent. After one minute, 450 μL $Na_2CO_3$ (20%, *w/v*) was added. The mixture was incubated for 120 min and absorbance of the resulting mixture was measured spectrophotometrically at 750 nm. Gallic acid was used as a reference standard and the results were expressed as milligram gallic acid equivalents (mg GAE)/g of extract.

2.2.4. Determination of Total Monomeric Anthocyanin Content (TAC)

TAC was determined according to [17] using the pH-differential method. Briefly, absorbance readings at 510 nm and 700 nm were made after dilution of extract in buffers solution, with pH values of 1.0 and 4.5, against distilled water. The calculation was based on Equations (1) and (2), respectively:

$$A = (A_{\lambda max} - A_{700})_{pH\ 1.0} - (A_{\lambda max} - A_{700})_{pH\ 4.5} \tag{1}$$

where Aλmax is the absorbance of the sample extract at 510 nm

$$Total\ monomeric\ anthocyanins \left(\frac{mg}{100g}\right) = A \times M_w \times Df \times \frac{1000}{\varepsilon \times l} \tag{2}$$

where $M_w$ (molecular weight) = 449.2 g/mole for cyanidin-3-glucoside; $Df$ = dilution factor; $l$ = pathlength in cm; $\varepsilon$ = 26,900 molar extinction coefficient in $L \times mole^{-1} \times cm^{-1}$ for cyanidin-3-glucoside; $10^3$ = conversion of g to mg. The results are expressed as mg cyanidin-3-glucoside per 100 g of onion dry matter. All analyses were performed in triplicate and the median values were calculated. The summary statistics of the chemical analyses are presented in Table 1. In the table below, Q1, Q2, and Q3 denote the quartiles. Q1 corresponds to the lowest 25% of numbers, Q2 ranges between 25.1% and 50% (up to the median), and Q3 corresponds to the range 50.1% to 75% (above the median).

**Table 1.** Summary statistics of Total Phenolic (TPC) and Total Monomeric Anthocyanin Content (TAC) values of the OS samples under study (n = 72).

| Parameter | Min | Max | Q1 | Q2 | Q3 | Mean |
|---|---|---|---|---|---|---|
| TPC (mg GAE/g) | 13 | 79 | 30 | 43 | 52 | 43 |
| TAC (mg Cyanidin/g) | 0.13 | 3.82 | 1.49 | 2.08 | 2.71 | 2.01 |

*2.3. Spectroscopic Characterization of the Dry OS*

In this section, we briefly present the methodological steps to develop an onion spectral library that would be used for assessing the total anthocyanin content. The building of a database for OS anthocyanins that utilizes their unique spectral signatures (combination of infrared bands) in specific spectral regions is described.

### 2.3.1. VNIR-SWIR Analysis

The VNIR-SWIR measurements of dry red OS powdered samples were performed using a PSR +3500 spectrometer (Spectral Evolution Inc., Lawrence, Massachusetts, USA) operating in the range 350 to 2500 nm. The measurements were performed using a contact probe to eliminate the effects of light scattering. Five spectra per powdered sample were recorded and averaged to obtain the corresponding reflectance spectral signatures. A Spectralon®panel with 99% reflectance was used to calibrate the spectrometer before the measurements.

### Spectral Preprocessing Techniques

Widely employed scatter-corrective and spectral-derivatization preprocessing techniques were applied to the VNIR-SWIR dataset to remove irrelevant information. In brief, (i) the reflectance spectra (REF) were converted into (ii) pseudo-absorbance spectra [log10(1/R)] (ABS), and (iii) transformed into a continuum removal method domain (CR). The Standard Normal Variate (SNV) was then applied to both REF and ABS values resulting to (iv) REF-SNV and (v) ABS-SNV datasets, respectively. The Savitzky–Golay method was applied to remove unwanted background noise from the spectra (vi) by calculating the first derivative (SG1), and in that case, (vii) combining with the SNV transformation (SG1-SNV) and also (viii) by calculating the second derivative (SG2), 11 data points of interval. Lastly, (ix) the detrend (DET) preprocessing method was used before data modelling. An overview of these techniques is presented by Rinnan et al. [18]. In total, nine different spectral datasets were produced.

### Machine Learning Modeling

The Conditioned Latin hypercube method (cLHS) [19] was used to split the onion VNIR-SWIR spectral data into calibration and validation datasets. According to the cLHS algorithm, the method searches the data based on heuristic rules combined with an annealing schedule. The proposed method is considered to be an effective way of replicating the distribution of the variables compared to a random sampling approach. The percentage of the number of onion samples to be allocated for the calibration dataset was determined as 75% of total dataset (54 out of 72), while the rest (18) were included in the validation set.

Each dataset of preprocessed spectra was modelled against TAC values using the following linear or non-linear regression algorithms, i.e., (i) partial least square regression (PLS); (ii) Random Forest (RF); (iii) Cubist; (iv) elastic net (ENET); (v) k-nearest neighbors (k-NN); and (vi) support vector machines for regression (SVM). In every method, a set of hyperparameters was selected as follows. The classical PLS algorithm [20], widely applied for multiple purposes in spectroscopic analysis, transforms the input factors' matrix into a series of latent variables (LVs) to maximize the covariance among the predictors and dependent variables. The number of optimum LVs was selected to range from 10 to 30. RF is an ensemble learning classifier [21] with good performance metrics in various spectroscopy studies. Tuning of hyperparameters included first the selection of a number of variables that can be sampled in each split of the tree analysis (6, 24) and then the value of the tree parameter (100, 250, 500, 1000, 1500). The rule-based Cubist algorithm [22] reduces a set of rules derived by a decision tree to define a linear regression model. Then, multiple rule-based models are combined (committees) and the final predictions are adjusted using known errors on the training set with a small number of neighbors for each unknown sample. Thus, predefined values for the number of committees (1, 10, 50, 100) and of neighbors (0, 1, 5, 9) were selected. The ENET method was also evaluated as an extension of linear regression that adds regularization penalties to the loss function during training. Next, the *k*-NN algorithm was used. This is an instance-based learning algorithm that utilizes a distance metric from the calibration dataset and predicts a testing pattern depending on a preselected number, k, of nearest neighbors. In our study, this value was optimized in a range from 0 to 25 closest k neighbors. Lastly, SVM for regression, as introduced by Drucker et al. [23], was evaluated. SVM is a non-parametric technique employing a kernel

function to map the initial predictors into a higher dimensional space. In this study, a radial basis function was utilized, while the C parameter values were optimized among (0.001, 0.01, 1, 10) to control the penalization of the residual errors.

A grid search on a five-fold cross-validation experiment for each analysis enabled the selection of the optimal hyperparameter values for model consistency. Table 1 in Appendix A shows the optimal hyperparameter values for each ML algorithm. In total, 54 calibration models were produced. In order to assess their performance for prediction of TAC in dry red OS, the root-mean-square error (RMSE, Equation (3)), the coefficient of determination ($R^2$, Equation (4)), and the Ratio of Performance to Interquartile Range (RPIQ; Equation (5)) values were compared. The equations used were as follows:

$$RMSE = \sqrt{\frac{\sum_{i=1}^{i=N}(y_i - \hat{y}_i)^2}{N}} \tag{3}$$

$$R^2 = 1 - \frac{\sum_{i=1}^{i=N}(y_i - \hat{y}_i)^2}{\sum_{i=1}^{i=N}(y_i - \bar{y})^2} \tag{4}$$

$$RPIQ = \frac{IQ}{RMSE} \tag{5}$$

where $y_i$ is the observed value and $\hat{y}_i$ is the predicted value of sample $i$, $N$ is the number of observations (Equation (3)), y is the mean of the observed values (Equation (4)), and IQ is the interquartile range (IQ = Q3 − Q1) of the observed values (Equation (5)). Q1 and Q3 denote the first and third quartile, respectively.

### 2.3.2. ATR-FT-MIR Analysis

ATR-FT-MIR spectra were acquired using a 6700 IR (Jasco, Essex, UK) spectrometer equipped with a DLaTGS detector, a high-throughput Single Reflection ATR with diamond crystal and complemented by the Spectra Manager software (Jasco, Essex, UK). For each spectrum, eight scans were accumulated in the absorbance mode and recorded at 4 cm$^{-1}$ resolution, covering a range from 4000 to 600 cm$^{-1}$. The spectrum was collected against a background obtained with a dry and clean cell and corrected by the ATR correction option of the software. Three spectra per powdered sample were recorded and averaged to obtain the corresponding spectrum before further preprocessing.

Spectral artifacts due to noise, baseline offset, and slope or light scattering were removed by the multiplicative signal correction method (MSC) and second order derivatization with the Savitzky–Golay method (11 data points of interval) [24]. The spectral data were mean-centered and further processed via Principal Component Analysis (PCA). PCA is an unsupervised technique that transforms a set of variables into a new set of composite variables, the principal components (PCs). PCA attempts to simplify the distribution of samples and identify the underlying factors that explain possible patterns of variable and sample correlations. For exploratory purposes, only principal components with eigenvalue >1.0 were considered useful, according to the Kaiser criterion [25].

### 2.4. Implementation

The statistical and regression analyses of the VNIR-SWIR datasets were performed utilizing the R programming language [26], with the caret package [27]. The commercial SIMCA 16.02 software (Umetrics, Sweden) was used for FT-MIR spectral analysis.

## 3. Results and Discussion

### 3.1. Chemical Analyses Data

The phenolic constituents of red onion skin are expected to exist primarily in bound form [5]. Nevertheless, the TPC values of red OS samples that represent mainly free soluble forms of phenolic compounds were found to range between 13 and 79 mg GAE/g. These values fall within typical ranges for the outer layers of brown-skin onion bulbs that have

previously been reported in literature [5,28], regardless of the geographical origin of the bulb or the extraction method.

The soluble phenolic extracts of OS samples were found to be rich in anthocyanins. In particular, the TAC values varied between 0.13 and 3.82 mg cyanidin per g. This result agrees with the findings reported in [5] and its references. However, it was observed that the samples originating from the Netherlands were far richer in monomeric anthocyanins (114.8–369.1 mg/100 g DW) than those from domestic sources (13.3–146.0 mg/100 g DW). A clear trend relating to geographical origin/retail chain was observed in the reference TAC values but not in evidence in TPC values. Whether the VNIR-SWIR and/or ATR-FT-MIR spectroscopic characterization of the samples would expose the same trend is intriguing.

### 3.2. VNIR-SWIR Exploratory Approach

It is accepted that bands at 1415–1512 nm, 1650–1750 nm, and 1955–2035 nm are mainly due to phenolic structure, according to the findings of Dykes et al. [29]. Similarly, in a research study about total anthocyanins in grape juice using NIR spectroscopy, it was found that the spectral range for these phenolic compounds was 1000–1183 nm [30]. Such bands along with those at around 1450 nm and 1930 nm, corresponding possibly to the O–H stretch and O–H band combination and the H–O–H deformation combination overtones of hydroxyl groups (e.g., due to water or starch) [16,31] were also evident in the near-infrared spectra of the dry OS samples under study.

As a general rule, the choice of an optimal preprocessing method depends on the characteristics of the dataset and the goal of the analysis [24]. In our study, the VNIR-SWIR spectra of OS samples as the original REF spectral values or as preprocessed spectra are illustrated in Figure 2. Visual assessment of the spectral signatures revealed no significant variation among the dry onion samples. Application of various preprocessing methods resulted in new feature spectral spaces by pronouncing different regions in the VNIR-SWIR spectrum and eliminating different effects. SG1 and SG2 emphasize the differences in the visible range and are more prominent to the SWIR region possibly because of greater overlapping of the bands. Similarly, the ABS (including also the SNV values) indicated larger variations than the REF at the first edge of the spectrum in the visible region (350–750 nm).

In a first exploratory approach to identify diagnostic patterns of sample distribution in the VNIR-SWIR, first derivative of the initial reflectance spectra (SG1) was analyzed via PCA. The analysis resulted in three PCs, the first of which (PC1) accounted for 82.8% of the total variance, the second (PC2) for 10.3% and the third (PC3) for 9.6%. The corresponding two-dimensional scoreplots verified that OS samples from domestic sources tended to be clustered separately from those originating from the Netherlands, mainly because of the higher PC2 score values of the latter (Appendix A, Figure A1). This result is quite promising for further modelling of TAC values given our previous observations. No other pattern could be recognized in the sample distribution among the 3-D scoreplot of the PCA model.

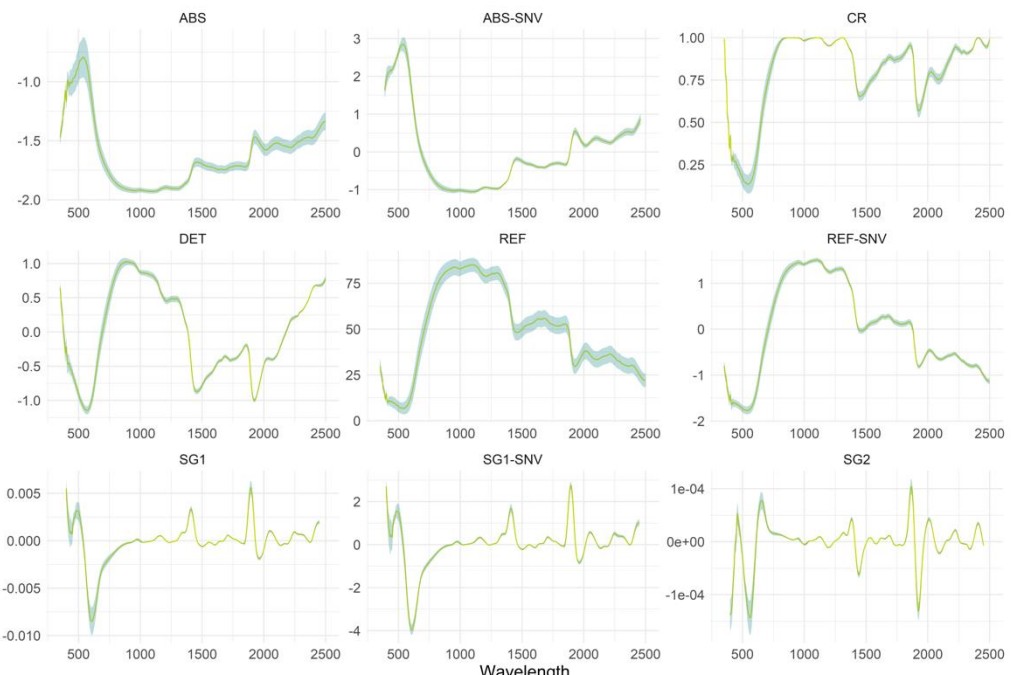

**Figure 2.** The original and preprocessed VNIR-SWIR spectra of OS samples. The abbreviations of the spectral pre-treatments are described in Section Spectral preprocessing techniques.

### 3.3. Monomeric Anthocyanins Prediction Based on the VNIR-SWIR Spectral Datasets

3.3.1. Performance of the ML Models

We first assessed six different ML models in different spectral datasets derived from the various pre-treatments to highlight the impact of the ML techniques in spectroscopic modelling. Overall, the proposed models have a valuable predictive performance ($R^2$ > 0.80, and RPIQ > 3). These findings are further illustrated in Figure 3. The results showed that spectral pre-treatments have increased the performance for most of the ML models, with the exception of the DET technique. Notably, modelling of the VNIR-SWIR onion spectral library with the pre-treatment of SG1-SNV and SG2 allowed more accurate predictions of TAC than other preprocessing techniques. A detailed comparison of the model performance obtained with various preprocessing techniques is also provided in Appendix B (Table A2).

We also tested the effectiveness of six ML models by comparing their performance metrics, as shown in Figure 4. In general, better predictive performance was achieved with more complex and supervised algorithms. The k-NN and RF algorithms were found to attain the best performance across all properties. They have enabled more robust predictions ($R^2$ > 0.80, RMSE < 0.53 and RPIQ > = 3.50). The results of the various models are reported graphically in Figure 4, in which we can visually compare their performances. The difference with the PLSR algorithm, one of the most commonly applied algorithms in food spectroscopy, is noticeable ($R^2$ = 0.81 and RPIQ = 3.48), the last having a larger RMSE (0.53). The Cubist and SVM results show lower predictive performance.

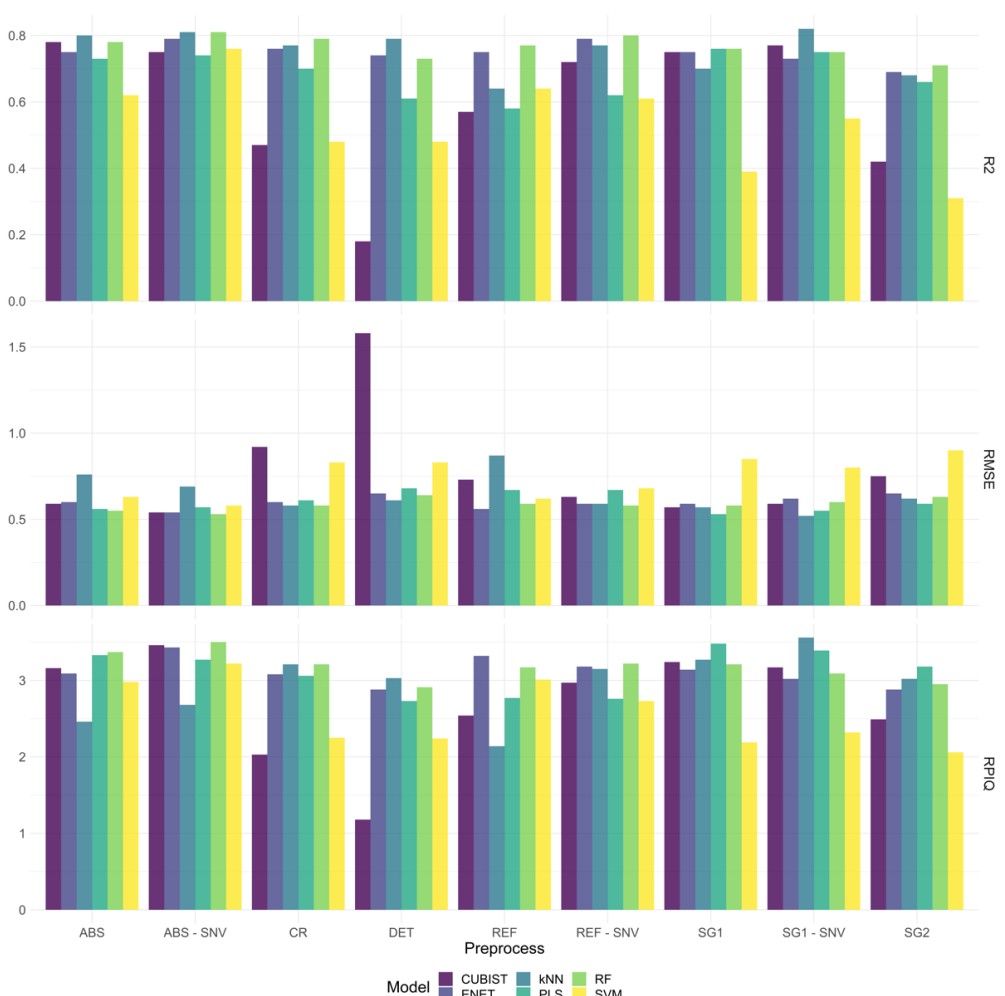

**Figure 3.** Comparison of the predictive performance metrics of the ML models using the various spectral sources.

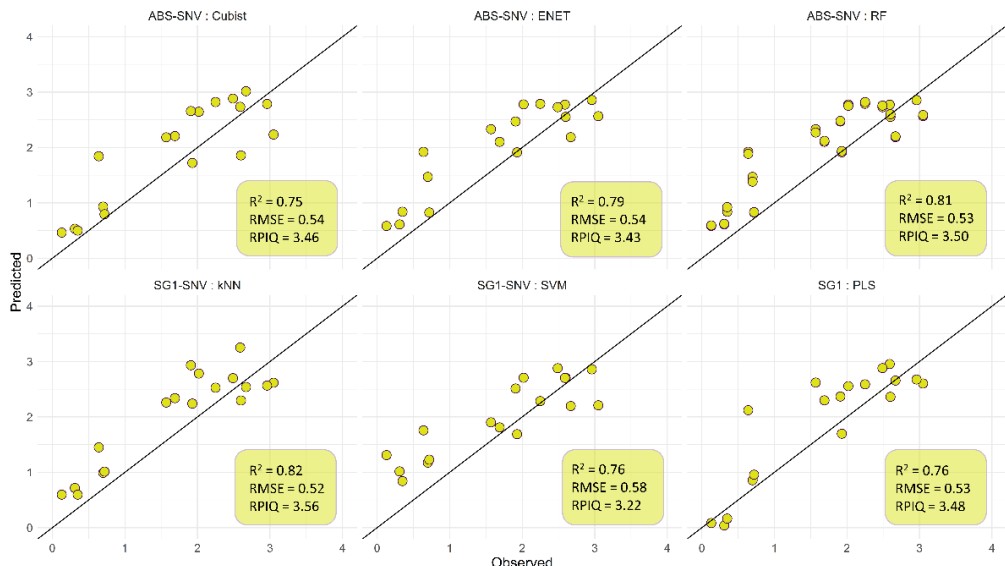

**Figure 4.** Independent validations set of the anthocyanins predictions from the ML models indicating the observed vs. the predicted values (black solid line represents the 1:1 line).

It should be highlighted that the selection of the ML model for regression analysis affects the prediction potential of VNIR-SWIR spectral data (Figures 3 and 4). The fact that k-NN and RF models presented better performance with smaller prediction errors than well-studied models in the domain of food spectroscopy (e.g., PLSR and SVM) may be a result of the efficiency of those algorithms to generate subsets with similar characteristics derived by different rules or the distance of the closest neighborhoods. Moreover, it was clearly demonstrated that the various spectral preprocessing techniques result in complementary information that enhances the predictive performance of the ML models compared to those produced with the raw reflectance recordings. Therefore, smoothing (SG1) and/or normalization of the dataset (SNV) should be prioritized in preprocessing steps.

An important aspect of the current study is the interpretability of the underlying models. By visualizing the relative importance of each band across all model-preprocessing combinations, it is possible to recognize those VNIR-SWIR spectral wavelengths that are more prominent in model construction (Figure 5). It is clear that two discrete spectral regions are important; the first one ranges from 400–1000 nm (VNIR), while the second one is in the range 1200–2500 nm (SWIR). Across nearly all models, VNIR has roughly two main spectral regions: one around 620–650 nm and the other at the beginning of the spectrum (380–390 nm), depicting, respectively, the characteristic red and purple color of the onion's layers due to the presence of anthocyanins. The finding that the upper SWIR part at 2100–2300 nm is practically related to aromatic C-H bonds provides valuable information.

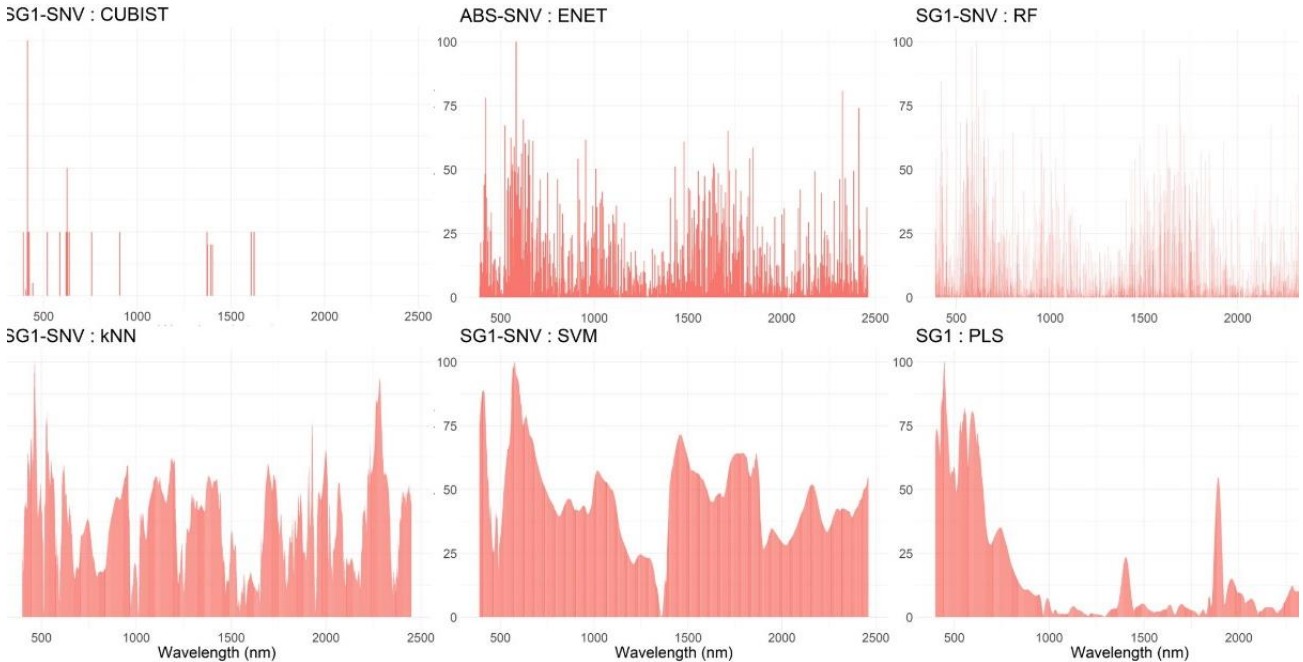

**Figure 5.** Relative importance of each wavelength in the VNIR-SWIR for the best ML model of TAC values. The spectral datasets used are: SG1-SNV, Savintzky–Golay 1st derivative with standard normal variate, ABS-SNV standard normal variate of absorbances, and SG1 1st derivative of reflectances. The ML algorithms are: ENET, elastic net; RF, random forest; k-NN, k-nearest neighbors; SVM, support vector machines for regression; PLS, partial least squares regression and the Cubist algorithm.

### 3.3.2. Exploring Shorter Diagnostic Regions in the VNIR-SWIR

New low-cost spectral devices available in the marketplace interest both researchers and end users to explore the potential of shorter spectral regions for anthocyanin content estimation. However, it is unclear if spectrometers operating solely at VNIR or SWIR could provide sufficient prediction accuracy. Therefore, focus was given to the spectral regions

between 400–1000 nm and 1350–2500 nm because they reflect more clearly spectral regions (as derived from the variable importance analysis) wavelength analysis (see Figure 5).

The *k*-NN model performance (see Figure 4) was tested in two shorter spectral ranges. Then, new rounds of modelling analysis were performed on two sub-sets of the SG1-SNV spectral dataset that corresponded to the selected regions. The results are shown in Table 2. It is clear that accuracy of VNIR-based prediction at 400–1000 nm ($R^2$ = 0.70, RMSE = 0.66, and RPIQ = 2.80) was better than the SWIR-based prediction at 1350–2500 nm ($R^2$ = 0.55, RMSE = 0.75 and RPIQ = 2.49), but not as high as of that corresponding to the combined spectral region (full spectrum).

**Table 2.** Predictive performance metrics of the best ML model using the full spectrum and limited spectral regions in VNIR (400–100 nm) and SWIR (1350–2500 nm).

| Spectral Range | $R^2$ | RMSE | RPIQ |
|---|---|---|---|
| Full spectrum | 0.82 | 0.52 | 3.56 |
| VNIR (400–1000 nm) | 0.70 | 0.66 | 2.80 |
| SWIR (1350–2500 nm) | 0.55 | 0.75 | 2.49 |

Chemometric analysis of the VNIR-SWIR spectra (350 to 2500 nm) resulted in satisfactory predictive performance of total anthocyanins content, selectively. The most important features for this purpose were a series of characteristic bands in the visible region of the spectra, mainly at 550–600 nm (at which these compounds mainly absorb), and in the range close to SWIR (2000–2300 nm) (Figure 5). Future studies could focus on the employment of shorter-range spectroscopic sensors for total anthocyanins in OSW to enable fast, low-cost analyses. This was also proposed recently regarding the evaluation of a Micro-Electromechanical systems spectral sensors for soil properties [32]. To computationally enhance the accuracy of prediction, more advanced chemometric approaches can also be employed. They may be combined, (predictions from single ML models developed using bootstrapped samples or the proposed ML algorithms developed via genetic stacking algorithms or even various spectral datasets after pre-processing) instead of relying solely on the best one via novel multi-input deep learning algorithms [33].

### 3.4. Identification of Phenolic-Group Diagnostic Bands in the MIR

The original FT-IR spectra and the accompanying spectral transformation (MSC, second derivative) of OS samples are shown in Figure 6.

In our study of dry OS powder, the characteristic amide-stretching bands of proteins (1550 and 1650 cm$^{-1}$) were not clearly evidenced in the FT-IR spectra. This finding is in line with earlier reports [34]. A weak valley at around 1560 cm$^{-1}$ revealed in preprocessed spectra signified a possible contribution from nucleic acid bases. Plant cell wall polysaccharides and other types of carbohydrates that are abundant in OS samples (e.g., fructooligosaccharides and pectic oligosaccharides) could be distinguished from some characteristic bands in the region between 1200 and 950 cm$^{-1}$ [35]. In addition, characteristic bands in the region 1800–1500 cm$^{-1}$ that are often assigned to esteried and non-esteried carboxyl groups of pectin molecules [34,35] were evidenced in the low frequency region.

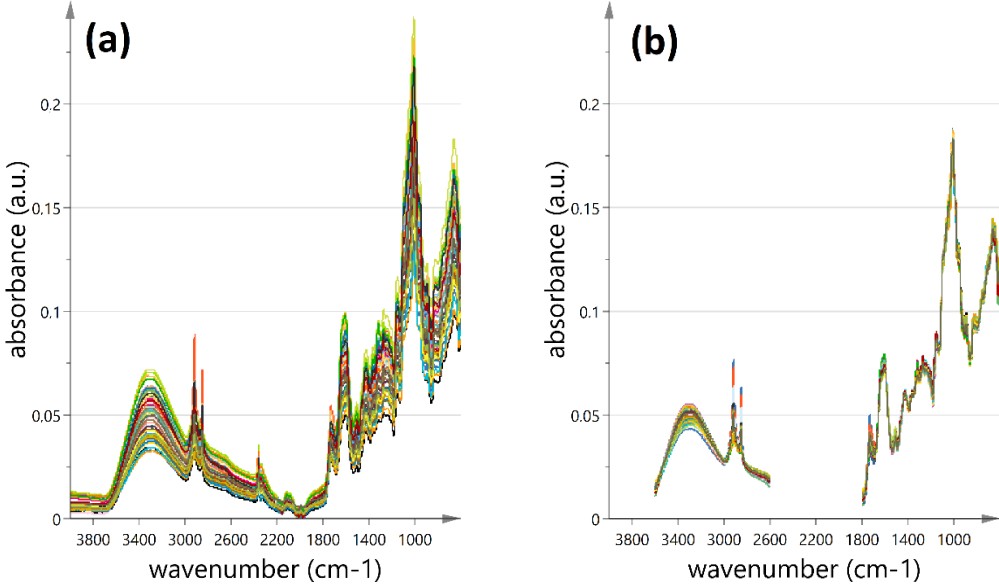

**Figure 6.** FT-MIR spectra of the dry onion skin samples under study (4000–600 cm$^{-1}$); (**a**) spectra without preprocessing; (**b**) 0th order curves after multi signal correction (MSC).

Table 3 provides an overview of the FT-IR spectral bands that were visually observed as peaks in the original spectra (0th order, after MSC) or as corresponding valleys in the 2nd derivative spectra (2nd order), respectively. The original peaks are clearly much better resolved after 2nd order derivatization of the spectra revealing a number of hidden bands that may carry diagnostic information. This was particularly evidenced in the region below 1000 cm$^{-1}$ but also between 1400 and 1600 cm$^{-1}$.

**Table 3.** Major bands shown as peaks in the 0th order and valleys in the 2nd order derivative FT-MIR spectra of dry OS samples and possible assignment.

| Sample Code | MIR Peaks | MIR Valleys | Functional Group Vibrations | Possible Identity |
|---|---|---|---|---|
| | 0th Order | 2nd Order | | |
| entry 1 | 3299.7 | | $v$(O-H), $v$(N-H) | carbohydrates, water, proteins |
| entry 2 | 2923.6 | 2917–2918 | $v_{as}$(C-H) | -CH3 and –CH2- alkanes |
| | 2850.3 | 2848–2850 | $v_{as}$(C-H) | CH3 or CH3 – Ar |
| | 1734.7 | 1734–1736 | $v$ (Ar-C = O) | aryl carboxylic acid monomers, e.g., hydroxybenzoic acids |
| | | 1717–1718 | $v$(Ar-C = O) | aryl ketones, aldehydes |
| | 1637–8 | 1636–1642 | $v$(Ar-C = O), $\delta$(H-OH) | aryl carboxylic acids and flavonoids, polygalacturonic acid peptides (amide I), water |
| | 1600–2 | | $v$(C = C aromatic), $v_{as}$(COO-) | flavonoids, polygalacturonic acid |
| | | 1561–1562 | | flavonoids, nucleic acid ring base |
| | | 1521 | $v$(C = C aromatic) | aryl carboxylic acids and flavonoids |
| | 1509 | 1503–1507 | $\delta$(C-H aromatic) | flavonoids |
| | | 1489 | $\delta$(C-H aromatic) | flavonoids, e.g., quercetin and cyanidin glucosides |
| | 1465 | 1471, 1464 | $\delta$(C-H aromatic) | aryl carboxylic acids and flavonoids |
| | 1423.2 | 1440, 1420 | $\delta$(C-H aromatic) & $v_s$(COO-) | aryl carboxylic acids, polygalacturonic acid esters |
| | | 1385 | $\delta$(O-H aromatic) | |

**Table 3.** *Cont.*

| Sample Code | MIR Peaks | MIR Valleys | Functional Group Vibrations | Possible Identity |
|---|---|---|---|---|
| | 0th Order | 2nd Order | | |
| | 1368 | 1363–1371 | δ(O-H aromatic) | flavonoids, e.g., catechin, polysaccharides |
| | | 1334 | ν(C-O), δ(C-H aromatic) | phenol |
| | 1321 | 1316–1319 | δ$_{sym}$ (-CH3 ), ν(C- H) | alkanes, alkenes, phenol or tertiary alcohol |
| | | | β(O-H) | |
| | 1271 | | | |
| | | 1200, 1230 | ν(Ar C-C-O), | phenols, carbohydrates |
| | 1152.3 | 1159–1166 | ν(C-O) | C–O–C glycosidic linkages of oligosaccharides |
| | | | ν(C-CO-C) | aliphatic ketones |
| | | 1101–1103 | ν (C-O), ν (C-C), ring | carbohydrates |
| | 1096 | | ν(C-C) | carbohydrates |
| | 1072 | | ν(C-OH) | oligosaccharides |
| | 1051 | 1049–1050 | | |
| | 1008.6 | | ν(C-O) | C–O–C glycosidic linkages of oligosaccharides, polysaccharides |
| | 987–988 | 985–986 | ω( = C-H), δ( = C-H) | polysaccharides, e.g., cellulose, aryl carboxylic acids, flavanols, |
| | | 972–973 | ω( = C-H), δ( = C-H) | polysaccharides, e.g., pectin, aryl carboxylic acids |
| | 954 | 951–952 | | carbohydrates, aryl carboxylic acids |
| | 892 | 891–894 | ω( = C-H) | substituted aromatic ring |
| | | 863–865 | ω( = C-H) | substituted aromatic ring |
| | 831–833 | 832–833 | ω( = C-H) | substituted aromatic ring |
| | | 756–758 | γ(C-H) | hydroxybenzoic acids |
| | | 717–718 | γ(C-H) | flavonoids |
| | | 706–710 | γ(C-H) | flavonoids, e.g., quercetin glycosides, epicatechin |
| | 665.3 | | | |
| | | 641–649 | | |
| | | 627–632 | | flavonoids, e.g., anthocyanin diglycosides |

[1] Based on [14,36–38]. ν—stretching, as—asymmetric, s—symmetric, β—in_plane bending, γ—out-of-plane bending, δ—scissoring or deformation, ϱ—rocking, ω—wagging.

For phenolic compounds, the two aromatic ring-related bands at around 1600 and 1640 cm$^{-1}$ were distinct in the spectra of dry OS powder [38]. These two bands were more clearly defined than those at 1185 and 965 cm$^{-1}$ possibly because of C-O-C and C-OH vibrations of phenols [38]. Given the copresence of polysaccharides and oligosaccharides in the test sample, straightforward assignment of the signals in the latter region is not possible. In a recent study about the potential of FT-IR and PCA to identify individual classes of phenols like flavonols, anthocyanins, and phenolic acids [39], the spectral bands between 1755 and 1400 cm$^{-1}$ and 1000 and 870 cm$^{-1}$ were highlighted as the most important. Based on our data, we suggest that stretching vibrations of carboxylic groups at around 1735 cm$^{-1}$ are more likely assigned to hydroxy-benzoic acid moieties that are present in OS as the major autoxidation products of quercetin. It has been reported that protocatechuic acid and 2-(3,4-dihydroxybenzoyl)-2,4,6-trihydroxybenzofuran-3 (2H)-one are formed during storage of the onions; during that period, enzymatic hydrolysis of quercetin glucosides to release the aglycone form proceeds in parallel with quercetin decomposition reactions [40]. Other conjugates formed due to auto-oxidation may also exist in relatively high amounts [41].

To examine further whether these tagged regions of the spectra have diagnostic value related to the total phenolic compound content of OS samples, the spectra of a group of totally 26 OS samples, representing the skin and the 1st inner layer of individual onion bulbs, were acquired and imported to the original dataset. Principal Component Analysis of the data from 0th and 2nd order derivative spectra in the regions 600–1800 cm$^{-1}$ resulted in 13 and 6 PCs, respectively. These PCs explained 99.2% and 75.7% of the total variance in each case. The analysis of 0th order data extracted eight PCs with very low eigenvalues (< 1) that explained almost 8% of total variance. This result verifies that a considerable amount of variance in the 0th order data is unique or not systematic and is omitted upon 2nd order derivatization. Both rounds of PCA showed that the two groups of samples could be distinguished on the PC1-PC3 scoreplot (Figure 7) on the basis of t3 score values.

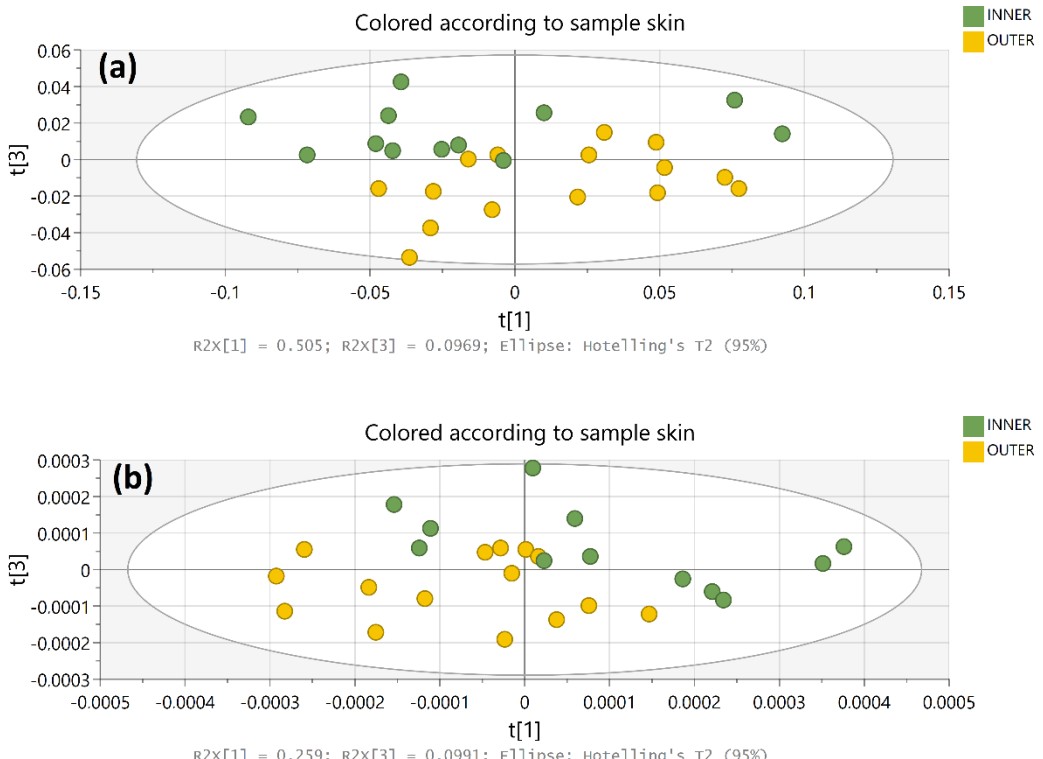

**Figure 7.** PCA scoreplots of (**a**) 0th order and (**b**) 2nd order derivative FT-MIR spectral data (600–1800 cm$^{-1}$) of dry OS. Different colors indicate outer and 1st inner layer samples from 15 bulbs.

The loading plots of the first and third PCs extracted from each round of PCA are shown in Figure 8. Spectral in the lower frequency region, e.g., at 613, 832, 948, 980 (possibly due to the phenolic ring structure), at 1012–1050 cm$^{-1}$ (sugars) along with that in the region between 1462–1472, 1500–1520, and at 1734 cm$^{-1}$ contributed more heavily to the t1–t3 score distribution ($p > \pm 0.6$) of these OS samples. Special attention was given to the observed variance between 600 and 1000 cm$^{-1}$ and 1400–1800 cm$^{-1}$ because it is expected to reflect more clearly differences in the composition of flavonoids and phenolic acid constituents [39]. New rounds of PCA on spectral data that corresponded only to shorter infrared regions revealed that the abundance of carboxylic acid groups remained a distinctive feature of the dry skin and first inner onion bulb layers (Figure 8). Further exclusion of variables between 900 and 1000 cm$^{-1}$ resulted in similar performance of the PCA model and verified (through corresponding loading plots) that the observed sample allocation is significantly affected by vibrations beyond that region (e.g., ether bonds in carbohydrates). Moreover, it made it possible to highlight that flavonoid ring-related bands around 600–650 cm$^{-1}$ and 1500–1560 cm$^{-1}$ are also important for the observed pattern among OS samples. Closer inspection of the FT-MIR spectral curves after 2nd

order derivatization revealed clear differences in the shape of the bands between 600 and 900 cm$^{-1}$ that might be partially attributed to skeletal vibrations of different flavylium ring substitution patterns. Considering that anthocyanins constitute a minor percentage of total flavonoids in the skin and outer layers of red onions [6] and the fingerprint region of the spectrum is dominated by highly overlapped signals, it is expected that these shorter-range FT-MIR bands do not assist in quantitative analyses. The FT-MIR data in the specific regions are promising for further exploratory evaluation considering mainly the potential for monitoring oxidation phenomena or other major sources of variance in the phenolic composition, but that is beyond the scope of this study.

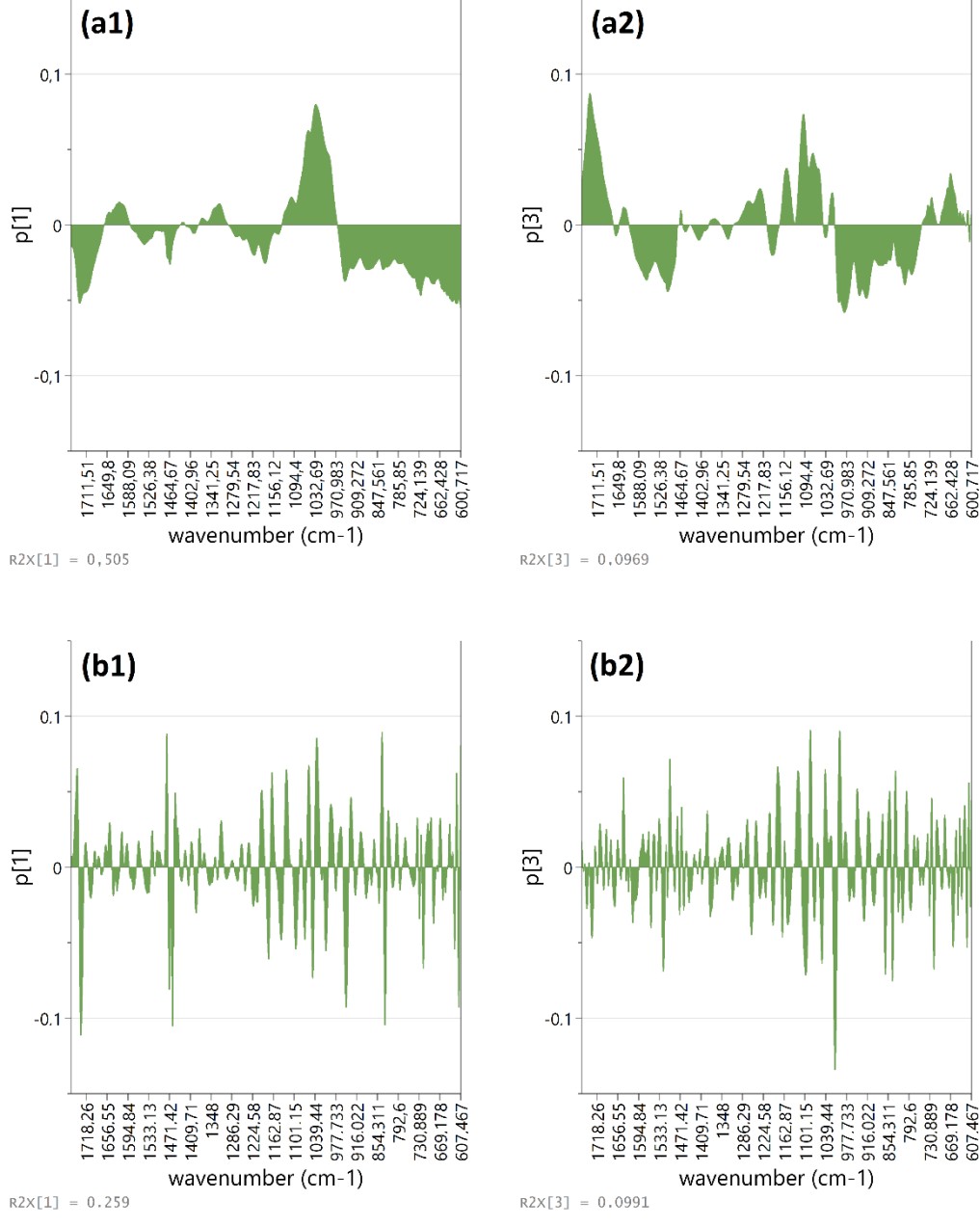

**Figure 8.** PCA loading plots showing the most important wavenumbers (600–1800 cm$^{-1}$) from (**a**) 0th order and (**b**) 2nd order derivative FT-MIR spectral data of dry OS for the formation of PC1 (**a1,b1**) and PC3 (**a2,b2**).

## 4. Conclusions

Given the heterogeneity of the OSW, which affects their chemical composition, the creation of reference databases is the most essential condition for building robust predictive models. Even though a relatively small OS sample set was used in the current study, the variance in total phenol and total anthocyanin contents of these samples was in the low-high ranges that are typically reported in literature. In the current study, we showed that VNIR-SWIR and FT-IR spectroscopic techniques could be deployed in routine quality control analyses of onion waste, especially in the evaluation of phenolic composition and more particularly in the assessment of the total anthocyanin content. Chemometric analyses of the data through various machine-learning techniques are indispensable for the identification of diagnostic bands across the visible near-to short-wave and mid-infrared regions. Above all, a k-NN model of 1st derivative spectra in the region of 350–2500 nm was the most powerful for the prediction of the monomeric anthocyanin content in dry red OS samples ($R^2$ = 0.82, RMSE = 0.52, and RPIQ = 3.56). The performance of the predictive model remained satisfactory when it was assessed in a shorter, more selective spectral range. This result supports the perspective for the potential uses of low-cost spectroscopic sensors in this field. The FT-IR spectral fingerprint was more informative about the inherent quality characteristics of OSW as it enables structural assignments. Overall, we suggest that non-destructive spectroscopic tools operating in the visible-near-short-wave and mid-infrared regions can be employed in real-time quality control of OSW if the spectral data are of high quality and well-demonstrated diagnostic value or predictive accuracy with regard to the audit target, e.g., anthocyanins content. Updating the reference onion skin spectral libraries and evaluation of the model performance are, thus, in progress.

**Author Contributions:** Conceptualization, N.T., S.A.O., G.Z. and I.M.; methodology, N.T., S.A.O., A.T.; software, K.K., S.A.O.; validation, N.T., S.A.O.; investigation, S.A.O. and A.T.; data curation, N.T., A.T. and S.A.O.; writing—original draft preparation, N.T., S.A.O., I.M.; writing—review and editing, S.A.O., G.Z., I.M.; visualization, N.T.; supervision, I.M. All authors have read and agreed to the published version of the manuscript.

**Funding:** This research received no external funding.

**Institutional Review Board Statement:** Not applicable.

**Informed Consent Statement:** Not applicable.

**Data Availability Statement:** The data presented in this study are available on request from the corresponding author (GM).

**Acknowledgments:** Authors would like to thank Interdisciplinary Agri-Food Center (KEAGRO), Aristotle University of Thessaloniki for providing the access to the equipment of the unit.

**Conflicts of Interest:** The authors declare no conflict of interest.

## Appendix A

An appropriate tuning of hyperparameters ensures the ML models' consistency. Thus, a grid search on a five-fold cross-validation experiment was conducted to select the optimal hyperparameters for ML model. The optimal set of hyper-parameters for each ML algorithm is presented in Table 1.

**Table 1.** Optimal hyper-parameters of ML models, selected by the grid search with the cross-validation approach.

| Pre-Treatment Methods | PLS | RF | | Cubist | | ENET | | k-NN | SVM | |
|---|---|---|---|---|---|---|---|---|---|---|
| | LVs | mtry | ntree | C | n | s | λ2 | k | C | sigma |
| ABS | 3 | 9 | 200 | 50 | 0 | 0.75 | 0.010 | 8 | 1 | 0.001 |
| ABS-SNV | 2 | 4 | 250 | 10 | 0 | 0.75 | 0.010 | 12 | 1 | 0.001 |
| CR | 2 | 19 | 150 | 100 | 9 | 0.75 | 0.005 | 6 | 1 | 0.001 |
| DET | 2 | 3 | 150 | 100 | 0 | 0.50 | 0.005 | 8 | 1 | 0.001 |
| REF | 2 | 20 | 150 | 1 | 0 | 0.75 | 0.010 | 13 | 1 | 0.001 |
| REF-SNV | 3 | 17 | 200 | 10 | 0 | 0.50 | 0.010 | 5 | 1 | 0.001 |
| SG1 | 2 | 13 | 250 | 10 | 5 | 0.50 | 0.010 | 6 | 1 | 0.001 |
| SG1-SNV | 2 | 2 | 150 | 50 | 0 | 0.50 | 0.010 | 3 | 1 | 0.001 |
| SG2 | 2 | 20 | 150 | 1 | 5 | 0.50 | 0.010 | 3 | 1 | 0.001 |

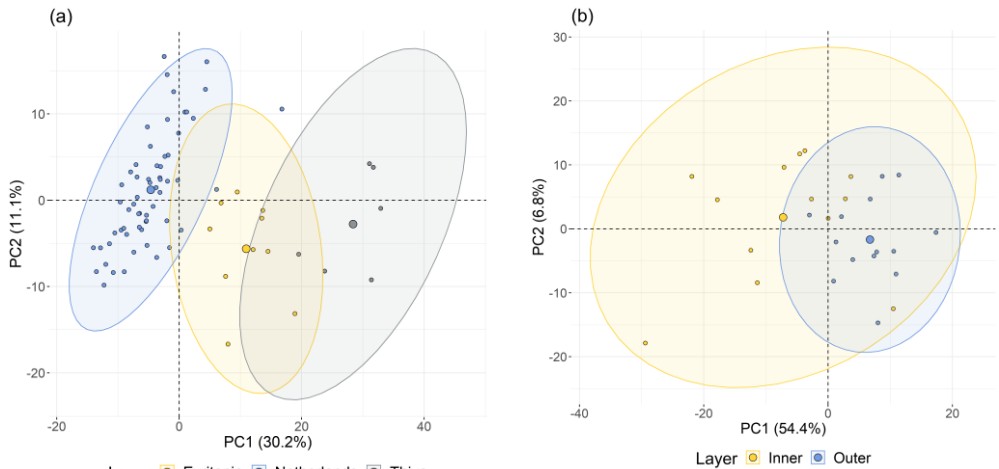

**Figure A1.** Scatter diagrams (**a**) of the first vs second principal components extracted from PCA of VNIR-SWIR spectral data of dry onion skin indicating the source of supply; and (**b**) of the first vs third principal components extracted from PCA of VNIR-SWIR spectral data of dry onion skin indicating the layer under study.

## Appendix B

**Table A2.** Accuracy results between the predicted and observed anthocyanins content values as derived by the application of the proposed ML models, in conjunction with the various pre-processing methods.

| Pre-Processing Techniques | PLS | | | RF | | | CUBIST | | | ENET | | | K-NN | | | SVM | | |
|---|---|---|---|---|---|---|---|---|---|---|---|---|---|---|---|---|---|---|
| | $R^2$ | RMSE | RPIQ | $R^2$ | RMSE | RPIQ | $R^2$ | RMSE | RPIQ | $R^2$ | RMSE | RPIQ | $R^2$ | RMSE | RPIQ | $R^2$ | RMSE | RPIQ |
| ABS | 0.57 | 0.67 | 2.76 | 0.76 | 0.58 | 3.17 | 0.57 | 0.73 | 2.54 | 0.75 | 0.56 | 3.32 | 0.64 | 0.87 | 2.13 | 0.63 | 0.61 | 3.01 |
| ABS-SNV | 0.73 | 0.55 | 3.32 | 0.78 | 0.55 | 3.37 | 0.78 | 0.58 | 3.15 | 0.74 | 0.60 | 3.09 | 0.79 | 0.75 | 2.46 | 0.62 | 0.62 | 2.97 |
| CR | 0.76 | 0.53 | 3.48 | 0.76 | 0.57 | 3.21 | 0.74 | 0.57 | 3.24 | 0.75 | 0.59 | 3.14 | 0.69 | 0.56 | 3.27 | 0.38 | 0.84 | 2.19 |
| DET | 0.66 | 0.58 | 3.17 | 0.70 | 0.63 | 2.94 | 0.42 | 0.74 | 2.49 | 0.69 | 0.64 | 2.87 | 0.65 | 0.61 | 3.03 | 0.31 | 0.90 | 2.05 |
| REF | 0.69 | 0.60 | 3.06 | 0.78 | 0.57 | 3.20 | 0.46 | 0.91 | 2.02 | 0.75 | 0.60 | 3.07 | 0.77 | 0.57 | 3.20 | 0.48 | 0.82 | 2.25 |
| REF-SNV | 0.62 | 0.67 | 2.76 | 0.80 | 0.57 | 3.22 | 0.72 | 0.62 | 2.97 | 0.79 | 0.58 | 3.17 | 0.76 | 0.59 | 3.14 | 0.60 | 0.68 | 2.72 |
| SG1 | 0.60 | 0.68 | 2.72 | 0.73 | 0.63 | 2.91 | 0.18 | 1.58 | 1.17 | 0.74 | 0.64 | 2.87 | 0.79 | 0.61 | 3.02 | 0.48 | 0.83 | 2.23 |
| SG1-SNV | 0.74 | 0.54 | 3.39 | 0.75 | 0.60 | 3.08 | 0.77 | 0.58 | 3.16 | 0.73 | 0.61 | 3.02 | 0.81 | 0.52 | 3.56 | 0.54 | 0.80 | 2.31 |
| SG2 | 0.74 | 0.56 | 3.27 | 0.81 | 0.53 | 3.49 | 0.74 | 0.53 | 3.46 | 0.79 | 0.54 | 3.43 | 0.81 | 0.69 | 2.68 | 0.76 | 0.57 | 3.22 |

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
