# Peer review of "Rapid Assessment of Anthocyanins Content of Onion Waste through Visible-Near-Short-Wave and Mid-Infrared Spectroscopy Combined with Machine Learning Techniques"

_sustainability, doi:10.3390/su13126588_

Round 1

Reviewer 1 Report

The paper is concentrated on  the methodological aspect of onion west evaluation. It is interesting subject of research in the scope of sustainable solutions. Presently, sustainable development of agriculture and industry is the one of the most important targets od the EU. Taking into consideration methodological aspect of the paper and detailed description of the methods, the paper was evaluated positively. Besides of general opinion, some detailed remarks were underlined in the review.

Author Response

Reply to the comments made by Reviewer #1

The paper is concentrated on the methodological aspect of onion wast evaluation. It is interesting subject of research in the scope of sustainable solutions. Presently, sustainable development of agriculture and industry is the one of the most important targets od the EU. Taking into consideration methodological aspect of the paper and detailed description of the methods, the paper was evaluated positively. Besides of general opinion, some detailed remarks were underlined in the review.

We would like to extend our gratitude to the reviewer #1 for appreciation of our work.

Author’s reply to specific remarks

Comment

The paper title: it is recommended to use full name of expressions “ VNIR-SWIR and FTIR” in the title or delate. Title should be clear.

Reply

Thank you for this remark. We have revised the title of the manuscript

Comment

the lack of information in the abstract about: the aim of the paper, the year of analysis, the place and country of the research, the main research results, the main conclusions.

Reply

We revised the abstract to clarify the aim, the methodology design, and the most important results-conclusions in terms of OSW quality control, based on reviewer#1 suggestions.

Comment

Please clarify in the abstract “VNIR-SWIR (350 – 2500 nm) and ATR-FT-IR (4000 -600 cm-1) spectra.”

Reply

Thank you for this remark. In the revised version we have clarified these terms. (lines xx-xx)

Comment

 Please clarify: “UV-Vis-based assays.”

Reply

We have accordingly revised the text.

Comment

The title of the table 1 is very long. I suggest to precise the title, while detailed information to add under the table.

Reply

The caption is changed based on your suggestions, while we provided a detailed description the text.

Comment

 The title of the figure 5, 7,8 is very long. I suggest to precise the title, while detailed information to add under the figures.

Reply

The titles were revised, as suggested

Comment

It it recommended to add some substantive conclusions in the scope of OSW. Presently, there were underlined only technical information about useful methods of OSW evaluation

Reply

We thank Reviewer#1 for this remark. Following his/her suggestion we have revised the abstract to better explain the practical importance of our methodology in terms of OSW quality control and management. This point is also addressed in our introduction and the conclusion part of the revised manuscript

Reviewer 2 Report

This work explores alternative methodologies to assess the content of Anthocyanins from red onions in situ.

This challenge is of high interest to the industry and to the valorisation of the food chain byproducts. 

In fact, my opinion about this paper was very concise because its a paper with an important and exhaustive approach to an alternative screening for the qualitative composition of bioactive compounds in onion, that was well written and with strong conclusions.

By applying machine learning together with analytical tools, the prediction of such profiles is extremely important for the food industry, therefore I think the paper is actually important and interesting.
What is more interesting about this, it that can be surely applied to other food sources rich in Anthocyanins (or other bioactives).

Reviewer 1 has already pointed out some typo errors and suggestions (namely the fact that the abbreviations need to be translated along the text) that I agree with and just want to add the following: 

Line 50 - esterified instead of esteried
Line 51 - colorants instead of colorant
Line 66 - remove "so far"

Author Response

Reply to the comments made by Reviewer #2

This work explores alternative methodologies to assess the content of Anthocyanins from red onions in situ.

This challenge is of high interest to the industry and to the valorisation of the food chain byproducts. 

In fact, my opinion about this paper was very concise because its a paper with an important and exhaustive approach to an alternative screening for the qualitative composition of bioactive compounds in onion, that was well written and with strong conclusions.

By applying machine learning together with analytical tools, the prediction of such profiles is extremely important for the food industry, therefore I think the paper is actually important and interesting.
What is more interesting about this, it that can be surely applied to other food sources rich in Anthocyanins (or other bioactives).

We sincerely thank reviewer#2 for considering our work important for the food industry and providing us positive feedback about the quality of methodology approach, comprehensiveness of conclusion points and potential for future applications. 

Author’s reply to specific remarks

Reviewer 1 has already pointed out some typo errors and suggestions (namely the fact that the abbreviations need to be translated along the text) that I agree with and just want to add the following: 

Comment

esterified instead of esteried

Reply

Thank you for your point. Following your suggestion, we corrected the words accordingly.

Comment

colorants instead of colorant

Reply

Thank you for your point. Following your suggestion, we corrected the word accordingly.

Comment

 remove "so far"

Reply

Thank you for your point. Following your suggestion, we removed these words.

Reviewer 3 Report

The paper addresses issues of Rapid Assessment of Anthocyanins Content of Onion Waste through VNIR-SWIR and FT-IR Spectral Analysis that is within the scopes of the journal. The novel contribution is the idea of the use ML-based exploratory approach to turn infrared spectroscopy into an operational tool for the assessment of the anthocyanins present in Onion Waste. The research paper describes a high-quality experimental design with the use of state-of-the-art spectral preprocessing techniques and methods for the documentation of rapid assessment of anthocyanins content of onion waste.

I think, at its present form, with improvements suggested in the attached file, the paper makes an acceptable case for publication.

It will be my omission if I do not propose to the authors that they should try to continue these kinds of experiments using more sophisticated methods, in terms of reflectance spectroscopy.

Author Response

Reply to the comments made by Reviewer #3

The paper addresses issues of Rapid Assessment of Anthocyanins Content of Onion Waste through VNIR-SWIR and FT-IR Spectral Analysis that is within the scopes of the journal. The novel contribution is the idea of the use ML-based exploratory approach to turn infrared spectroscopy into an operational tool for the assessment of the anthocyanins present in Onion Waste. The research paper describes a high-quality experimental design with the use of state-of-the-art spectral preprocessing techniques and methods for the documentation of rapid assessment of anthocyanins content of onion waste.

I think, at its present form, with improvements suggested in the attached file, the paper makes an acceptable case for publication.

It will be my omission if I do not propose to the authors that they should try to continue these kinds of experiments using more sophisticated methods, in terms of reflectance spectroscopy.

We sincerely thank reviewer#3 for pointing out our idea to turn infrared spectroscopy into an operational screening tool for the onion waste quality assessment and considering our experimental design as of high-quality. We fully agree that other types of reflectance measurements could be valuable in this field and need thorough investigation.

Author’s reply to specific remarks

We have adopted the corrections made in the annotated text by reviewer#3.

Moreover, we carefully checked for other spelling/typographical errors throughout the text and made additional corrections to improve the quality of our figures, legends etc, where needed. All these changes can be tracked in the file with the revised version of the manuscript.